# Ductus Venosus Agenesis as a Marker of Pallister–Killian Syndrome

**DOI:** 10.3390/medicina55070374

**Published:** 2019-07-15

**Authors:** María Victoria Lapresa Alcalde, Ana María Cubo, María Carmen Martín Seisdedos, Javier Cortejoso Hernández, María José Doyague Sanchez, José María Sayagués

**Affiliations:** 1Department of Obstetrics and Gynecology, University Hospital of Salamanca and IBSAL, Paseo San Vicente 58-182, 37007 Salamanca, Spain; 2Department of Genetics, University Hospital of Salamanca, Paseo San Vicente 58-182, 37007 Salamanca, Spain; 3Department of Obstetrics and Gynecology, University Hospital of Valladolid, Calle Rondilla Sta. Teresa, s/n, 47010 Valladolid, Spain; 4Department of Hematology, University Hospital of Salamanca and IBSAL, Paseo San Vicente 58-182, 37007 Salamanca, Spain

**Keywords:** ductus venosus agenesis, Pallister–Killian syndrome, array-CGH, microarrray, prenatal diagnosis

## Abstract

The ductus venosus (DV) is a shunt that allows the direct flow of well-oxygenated blood from the umbilical vein (UV) to the coronary and cerebral circulation through the foramen ovale. Its agenesis has been associated with chromosomal abnormalities and rare genetic syndromes, structural defects, intrauterine growth restriction (IUGR) and even antepartum fetal demise. Pallister–Killian Syndrome (PKS) is a rare sporadic disorder with specific tissue mosaic distribution of an extra 12p isochromosome (i(12p)). Its main clinical features are moderate to severe intellectual disability/neuromotor delay, skin pigmentation abnormalities, typical facial appearance, variable association with multiple congenital malformations and epilepsy. Though prenatal findings (including congenital diaphragmatic hernia, ventriculomegaly, congenital heart disease, polyhydramnios, and rhizomelic shortening) have been described in literature, prenatal diagnosis is difficult as there are no associated identification signs no distinctive or pathognomonic signs, and some of these malformations are hard to identify prenatally. The tissue mosaicism linked to this syndrome and the decrease of the abnormal clone carrier of the i(p12) after successive trypsinizations of cultured cells makes the diagnosis even more challenging. We present the case of a 27.5 weeks pregnant woman with a fetal ductus venosus agenesis (DVA) as the main guide marker. To our knowledge this is the first case published in literature reporting a DVA as a guide sign to diagnose a complex condition as Pallister–Killian syndrome. We also underscore the key role of new genetic techniques as microarrays to avoid misdiagnosis when only a subtle sonographic sign is present in complex conditions like this.

## 1. Introduction

The ductus venosus (DV) is a shunt that allows the direct flow of well-oxygenated blood from the umbilical vein (UV) to the coronary and cerebral circulation through a preferential passage across the foramen ovale. It is routinely assessed in the first trimester screening for fetal aneuploidies [1], in the second trimester anomaly scan [2] and it is useful to establish the stage in case of fetal growth restriction [3]. The reported prevalence of ductus venosus agenesis (DVA) is very variable, ranging from 1/2532 [4] to 1/167 [5] depending on the analyzed series. The agenesis of this relevant shunt has been associated with chromosomal abnormalities and rare genetic syndromes, structural defects, intrauterine growth restriction (IUGR) and even antepartum fetal demise [1,5,6,7].

Pallister–Killian Syndrome (PKS, OMIM 601803) is a rare sporadic disorder with specific tissue mosaic distribution of an extra isochromosome 12p. This results in the existence of four copies of the short arm of chromosome 12 in the affected cells. PKS is an unusual genetic syndrome, with an estimated prevalence of 5.1 cases per million live births [8], and it is severe, as it usually involves serious mental retardation. Its main postnatal clinical features are moderate to severe intellectual disability/neuromotor delay, skin pigmentation abnormalities, typical facial appearance, variable association with multiple congenital malformations and epilepsy [8,9,10,11]. Though prenatal findings (including congenital diaphragmatic hernia, ventriculomegaly, congenital heart disease, polyhydramnios, and rhizomelic shortening) have been described in the literature, prenatal diagnosis is difficult as there are no distinctive or pathognomonic signs, and some of the associated malformations are hard to identify prenatally [9,10,12,13]. Regarding serum markers, neither B-HCG nor PAPP-A have demonstrated to be useful to diagnose this syndrome [9]. Prenatal diagnosis of PKS is also a challenge due to difficulties in detecting the extra iso (12p) chromosome, the variable level of mosaicism and the rapid decrease of the supernumerary marker isochromosome during amniocyte sub-culturing [9,13,14]. Here, we report a case of prenatally diagnosed PKS with DVA as the main sonographic guide sign. To our knowledge, this is the first described in the literature. Considering the diagnostic difficulties of this syndrome, as there are no distinctive or pathognomonic markers, the presence of a possible new marker could be useful for the diagnosis of the disease.

## 2. Case Report

We report the case of a 36-year-old woman, who was 27.5 weeks pregnant and had no antecedents of interest or consanguinity with her partner. The patient was referred to us from a private clinic for ductus venosus agenesis (DVA), polyhydramnios and small fetal stomach with suspicion of oesophageal atresia. The patient was controlled outside of our unit until that moment; however, her first trimester combined screening results were low-risk. The study was reviewed and approved by the University Hospital of Salamanca Institutional Review Board (no. PI 201 8 03 205; date of approval 18 March 2018). The patient referred in the case report participant provided informed written consent.

Our exploration showed a bifurcation of the umbilical vein, giving rise to a normal hepatic-portal portion and another varicose portion of 5.2 mm, which ran intrahepatic, describing a curved path in the form of an intrahepatic "C" (Figure 1A). This joined to the suprahepatic veins and drained into the inferior vena cava (IVC), giving rise to an umbilical-IVC portosystemic shunt. The fetal stomach was normal, though a slight polyhydramnios was present (27 Amniotic Fluid Index). Echocardiography showed a cardiac axis displaced to the left, with the wall of the right ventricle thickened, especially at the level of tricuspid valve implantation. The Tei-Index was 0.59 (>p95 for gestational age at the expense of isovolumetric relaxation time), which indicated mild diastolic dysfunction. Estimated fetal weight (EFW) was in the 77th percentile, with fetal biometry parameters as follows: Biparietal Diameter (BPD) = 70.8 mm (p63), Head Circumference (HC) = 287.7 mm (>p98), Abdominal Circumference (AC) = 241.4 mm (p64) and Femur Length (FL) = 53.8 mm (p59) (Figure 1B). The rest of the examination was normal.

Amniocentesis was offered, requesting array-CGH and molecular study through a massive panel sequencing of 16 genes related to Noonan Spectrum Disorders (RASopathies), finding no pathogenic mutations in the sequences analyzed.

The analysis by array-CGH of the DNA extracted from uncultivated amniocytes obtained a result of arr (hg19) 12p13p11 (222,688-34,345,726) × 4 and male genomic pattern (Figure 2A). A gain of chromosomal material was detected in the entire short arm of chromosome 12, compatible with Pallister–Killian Syndrome (PKS) (Figure 2B). After obtaining the result, we proceeded to perform karyotype from a sample of amniotic fluid kept in culture in the Genetics Service to visualize the 12p isochromosome and confirm this finding. To our surprise, however, 3 weeks after its extraction, the result of the conventional cytogenetic showed a normal karyotype, probably due to clonal selection of normal diploid cells over the tetrasomal ones during the culturing process. The couple was informed of these results, deciding to terminate the pregnancy. Unfortunately, termination of pregnancy was carried out in a different center and no autopsy report was available.

## 3. Discussion

Pallister–Killian syndrome (PKS), also known as 12p mosaic tetrasomy, is a multisystemic disease whose prenatal diagnosis is usually an incidental finding when performing a karyotype in cases of increased nuchal translucency or fetal anomaly [10]. According to literature, the most frequent prenatal findings are polyhydramnios (37%), long bone shortening (34%), brain anomalies (20%), congenital diaphragmatic hernia (27%) fetal macrosomia (7%) cardiac malformations (16%), genitourinary malformations (10%), polydactyly and feet anomalies (19%) and gastrointestinal malformations (10%) [9]. Analyzing each of these groups, the most frequent brain malformations reported were ventriculomegaly (71%), Dandy Walker malformation (6%) and cerebellar anomalies (12%), whilst regarding cardiac malformations the most frequent ones were left ventricular hypoplasia (29%) and anomalies in aortic and pulmonary valves (28%), among others [9]. Other more sporadic anomalies are: fetal hydrops (6%), cystic hygroma (7%), increased nuchal translucency (9%) and presence of sacral appendix (3%) [9,10]. The presence of alterations in the fetal profile (small nose, thin upper lip with thick and protruding lower lip) [12] has also been related to this syndrome as well as BPD and HC measurements tending to be above the mean while fetal femur growth is under normal percentiles for gestational age [9]. There are, however, no pathognomonic sonographic indicators. In our case, only a slight polyhydramnios (AFI = 27), ductus venous agenesis (DVA) and mild diastolic dysfunction were observed. Fetal profile was normal and fetal biometry was slightly increased (p77) with an oversized HC (>p98), but not reaching macrosomia threshold. Femur size was normal at 27.5 weeks (p59), which is inconsistent with rizomelia. The rest of the fetal parameters were normal, with DVA being the main marker that led to performing a genetic study and, thus, to the diagnosis.

Although prenatal diagnosis of PKS can be performed by chorionic villus sampling (CVS), amniocentesis or cordocentesis, amniocentesis seem to be the best method [15] with detection rates reported ranging from 40 to 75% for CVS and 78–95% for amniocentesis [9,13,14]. Due to the high risks associated to cordocentesis, it is not a first diagnostic option. The conventional cytogenetic diagnosis of PKS is especially difficult due to the peculiarity of tissue-dependent mosaicism that it presents; in fact, although it is common to identify the 12p isochromosome in fibroblast cultures, it is rarely identified in blood lymphocytes and cytogenetic discordances have been described between placental and fetal tissues [15,16]. This variability could reflect a selective proliferative advantage of normal diploid cells over the tetrasomal cells during embryogenesis [14,17], although variable tissue-dependent mechanisms have also been postulated for the selection of cell lines in vivo or in vitro. In vivo, the loss of i(12p) has been shown with the aging of patients in bone marrow, fibroblasts and lymphocytes, whilst in vitro, a progressive decrease of the abnormal clone carrier of the isochromosome p12 has been observed after successive trypsinizations of cultured cells, independently of the tissue that is analyzed [17]. This loss of the abnormal clone in successive cultures has special importance from the point of view of prenatal diagnosis, especially in those cases in which the cells grow slowly in the culture and repeated trypsinizations are necessary, since this can lead to false diagnoses. If PKS is prenatally suspected the laboratory should be advised in order to not use long-term cultures and/or in order to perform an array-CGH analysis. FISH analysis by using alpha-satellite or 12 fluorescent probes on interphase nuclei of amniotic cells can increase the i(12p) detection rate [9,14].

The DVA is the result of a failure in the formation of the critical anastomosis between the portal-umbilical venous system and the hepatic-systemic venous system; there being no direct communication between the UV and the right heart. The blood is therefore diverted through an aberrant vessel. Two different routes for the return of the UV have been described in these fetuses, extrahepatic drainage or portosystemic shunt (direct connection of the UV to the iliac vein, to the ICV, to the right atrium (RA), to the suprahepatic veins or even to the coronary sinus) and intrahepatic drainage or hepatic umbilical shunt (connection of the UV to the portal sinus) [5]. The portosystemic shunt seems to be associated with a worse perinatal prognosis and a higher rate of associated structural and genetic defects [1].

The finding of a DVA should be complemented with a detailed study of the fetal anatomy, since the association to both cardiac and extracardiac malformations is described in more than 80% of cases, including septal defects, hypertrophic cardiomyopathy, facial clefts, intestinal malrotation and additional anomalies of the venous system. The study should also include a fetal genetic study, since genetic syndromes such as trisomies, Turner and Noonan syndromes are also described [1,18]. Until now, the traditional genetic study of these fetuses has been by conventional karyotyping techniques. However, such an approach has several limitations, mainly related to the need for obtaining metaphases, usually after an in vitro cell culture period. The greater expansion of molecular genetic diagnostic techniques, such as array-CGH, allows the diagnosis of complex syndromes that may not be detected using only the conventional karyotype. Following the recommendations of the consensus conferences for the use of the microarrays in prenatal diagnosis [19,20], which establish the preferred use of this techniques when the echographic findings suggest non-specific fetal anomaly of a complete syndrome, complementing with karyotype either in a second time or in parallel, a prenatal array was requested in our case, with the result of PKS. After this finding, a karyotype was requested as a second step, to complete the study; however, this finding could not be confirmed in the karyotype due to mosaic nature of the syndrome and isochromosome loss during in vitro culture. In our case, none of the typical clinical manifestations reported in the literature of Pallister–Killian syndrome was present, except a slight polyhydramnios (Amniotic Fluid Index = 27) [10,12]. We cannot predict if with time differences between HC and FL would become more evident and the FL would fall in lower percentiles as reported in literature, since the papers that highlight the PKS peculiar growth pattern report it in later gestational ages. However, our patient decided to legally terminate the pregnancy and no further follow-up was possible. Thus, the guide sign for the prenatal diagnosis was DVA. This is the first case to date reported in the literature, since this finding is not reflected in any of the reviews analyzed. Considering that DVA is a subtle sign that can be associated with complex syndromes, we suggest that its presence in conjunction with polyhydramnios and fetal weight above the mean or other minor signs should alert for PKS and encourage the search of genetic conditions associated to this finding, using the microarrays as the first choice diagnostic technique, complementing it with conventional karyotype if necessary to improve diagnosis. It is even more important in rare genetic conditions such as PKS, as the rapid loss of the i(12p) in the course of amniocyte subculturing could lead to misdiagnosis.

## Figures and Tables

**Figure 1 medicina-55-00374-f001:**
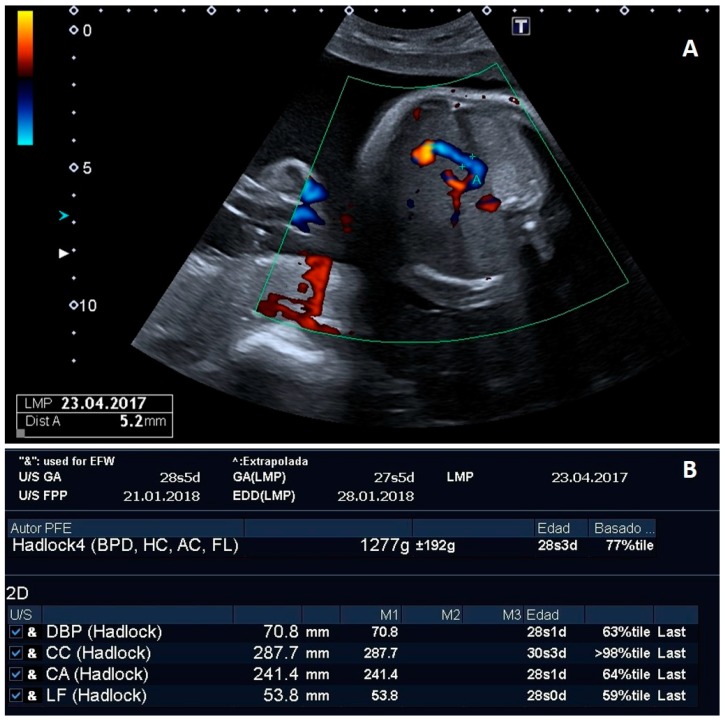
Ultrasonographic image showing “C-shaped” intrahepatic varicose portion of the umbilical vein without fetal anomalies (**A**) and fetal biometry report showing estimated fetal weight in 77th percentile as well as an oversized head circumference (>98th percentile) and normal femur length (59th percentile) (**B**).

**Figure 2 medicina-55-00374-f002:**
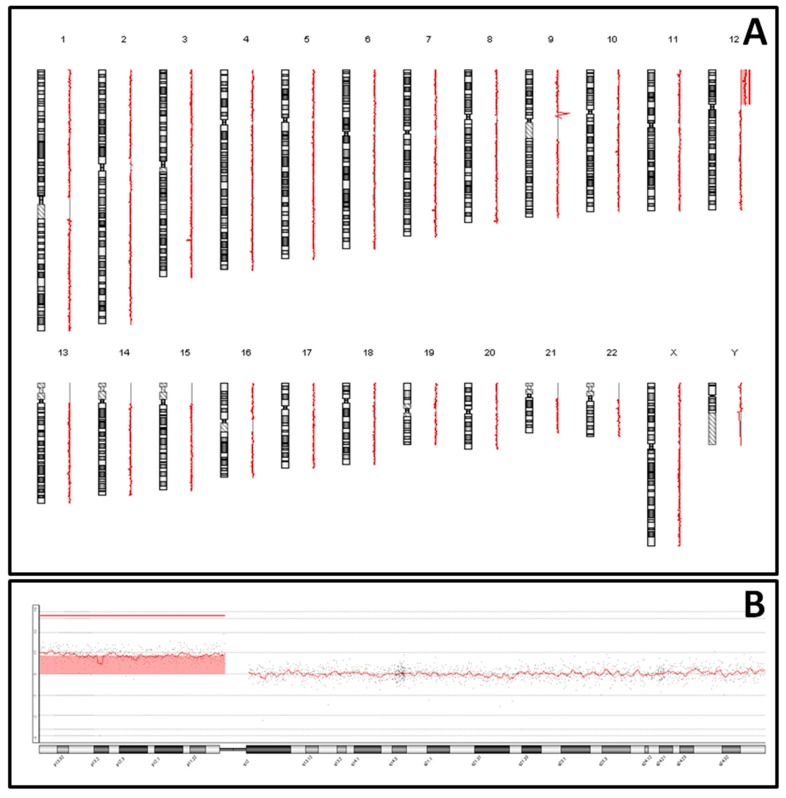
Copy number alterations in fetal genome assessed by array-CGH (**A**) and microarray plot for a clinically significant 34.12-Mb gain at 12p13p11 (**B**).

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
