# Peer review of "Ductus Venosus Agenesis as a Marker of Pallister–Killian Syndrome"

_medicina, 2019, doi:10.3390/medicina55070374_

Reviewer 1 Report

lines 19-20

There is an inconsistency between these two phrases. In the first one you state that PKS is associated with "multiple and varied malformations" but in the second phrase you state on contrary that "there are no associated identification signs". Did you mean that there are no distinctive or pathognomonic signs?

line 24

In the main text you describe not only DVA but also small fetal stomach observed in other clinic and polyhydramnios. In the abstract you indicate DVA as the only one sonographic marker which is misleading.

line 34

DV is routinely screened in the first trimester and what about second trimester?

line 41

an unusual

lines 43-44

There are several anomalies associated with PKS and described in the literature. Herein one can understand that PKS fetuses present no anomalies on ultrasound examination. 

lines 45

Like in the line 24: other features were: small fetal stomach and polyhydramnios

line 72-73

Isochromosome is a cytogenetic definition. 12p gain in array-CGH is compatible with PKS diagnosis but not necessarily with isochromosome 12p.

line 79

Do you have autopsy report? That needs to be written up. Can you confirm or exclude esophageal atresia?

lines 136-137

This finding could not be confirmed in the karyotype due to mosaic nature of the syndrome and isochromosome loss during in vitro culture

138

There is polyhydramnios

Author Response

REVIEWER #1: 
Comment 1.- lines 19-20: There is an inconsistency between these two phrases. In the first one you state that PKS is associated with "multiple and varied malformations" but in the second phrase you state on contrary that "there are no associated identification signs". Did you mean that there are no distinctive or pathognomonic signs? 
Answer to comment 1.- In accordance with the reviewer's note, we have amended the paragraph, which now reads as follows: “no distinctive or pathognomonic signs, and some of these malformations are hard to identify prenatally”. 
Comment 2.- line 24: In the main text you describe not only DVA but also small fetal stomach observed in other clinic and polyhydramnios. In the abstract you indicate DVA as the only one sonographic marker which is misleading. 
Answer to comment 2.- We fully agree with the reviewer's comment. This paragraph may sound confusing and it has been rewritten to read as follows: “We present the case of a 27.5 weeks pregnant woman with a fetal ductus venosus agenesis (DVA) as the main guide marker”. 
Comment 3.- line 34: DV is routinely screened in the first trimester and what about second trimester? 
Answer to comment 3.- Information about DV ultrasound screening and usefulness has been added following the reviewer’s comment. It is routinely assessed in the first trimester screening for fetal aneuploidies, in the second trimester anomaly scan and it is useful to establish the stage in case of fetal growth restriction. 
Comment 4.- line 41: an unusual 
Answer to comment 4.- We thank the reviewer for noticing this grammatical mistake. It has been corrected in the text. 
Comment 5.- lines 43-44-45: There are several anomalies associated with PKS and described in the literature. Herein one can understand that PKS fetuses present no anomalies on ultrasound examination. 
Answer to comment 5.- Information about the anomalies associated with PKS has been added in the discussion section of the new revised version of the manuscript; though prenatal findings (including congenital diaphragmatic hernia, ventriculomegaly, congenital heart disease, polyhydramnios, and rhizomelic shortening) have been described in the literature, prenatal diagnosis is difficult as there are no distinctive or pathognomonic signs, and some of the associated malformations are hard to identify prenatally. Regarding serum markers, neither B-HCG nor PAPP-A have demonstrated to be useful to diagnose this syndrome. Prenatal diagnosis of PKS is also a challenge due to difficulties in detecting the 12p gain, the variable level of mosaicism and the rapid decrease of the supernumerary marker isochromosome during amniocyte sub-culturing. Here, we report a case of prenatally diagnosed PKS, with DVA as the main sonographic guide sign. To our knowledge, it is the first described in the literature. Considering the diagnostic difficulties of this syndrome, as there are no distinctive or pathognomonic markers, the presence of a possible new marker could be useful for the diagnosis of the disease. 
Comment 6.- lines 45: Like in the line 24: other features were: small fetal stomach and polyhydramnios 
Answer to comment 6.- We thank the reviewer for this comment, which has been addressed accordingly (see answer to comment 2). 
Comment 7.- line 72-73: Isochromosome is a cytogenetic definition. 12p gain in array-CGH is compatible with PKS diagnosis but not necessarily with isochromosome 12p. 
Answer to comment 7.- This sentence has been modified following the reviewer’s comment. “A gain of chromosomal material was detected in the entire short arm of chromosome 12, compatible with Pallister-Killian Syndrome (PKS)”. 
 Comment 8.- line 79: Do you have autopsy report? That needs to be written up. Can you confirm or exclude esophageal atresia? 
Answer to comment 8.- Unfortunately, termination of pregnancy was carried out in a different centre and no autopsy report was available. It has been reported in the new revised version of the manuscript. 
 Comment 9.- lines 136-137: This finding could not be confirmed in the karyotype due to mosaic nature of the syndrome and isochromosome loss during in vitro culture. 
Answer to comment 9.- We thank the reviewer for reformulating this sentence, which sounded confusing and unclear. It has been corrected following his/her advice. 
Comment 10.- line 138: There is polyhydramnios 
Answer to comment 10.- We agree with the reviewer's point. Following his/her advice, the paragraph has been corrected to read as follows: In our case, none of the typical clinical manifestations reported in the literature of Pallister Killian syndrome was present, except a slight polyhydramnios (Amniotic Fluid Index=27): the guide sign for the prenatal diagnosis was DVA. 

Reviewer 2 Report

Major Comments:

Although the manuscript does not provide any specific new inside the prenatal cytogenetic diagnostic approach for PKS, it’s always useful reporting on a new sign associated with a such rare as well as likely underdiagnosed condition as PKS.

I would suggest authors to provide a more specific background on prenatal PKS findings, carefully studying all the dedicated literature (there are some recent studies not even cited in the text and some data reported from literature are not properly updated).

It would be interesting also to report a fetal post-mortem examination if an autopsy has been performed in order to get confirmation of prenatal findings detected by ultrasonography.

Overall the manuscript is well-written, however it may benefit of some polishing by a native English collegue. 

Minor Comments:

Abstract

Line 19: “Pallister-Killian Syndrome (PKS) is a rare sporadic disorder with specific tissue mosaic

distribution of an extra 12p isochromosome (i(12p)) that associates severe mental retardation as well as multiple and varied malformations”. I would say that the main clinical features of PKS are moderate to severe intellectual disability/neuromotor delay, skin pigmentation abnormalities, typical facial appeaence variable associated with multiple congenital malformations and epilepsy

Line 20: “Prenatal diagnosis is difficult as there are no associated identification signs”

Prenatal data from a cohort of 114 PKS probands have been recently published (Am J Med Genet A. 2018 Dec;176(12):2575-2586. doi: 10.1002/ajmg.a.40499. Epub 2018 Oct 5) providing guidelines for early recognition of the distinctive prenatal profile and consideration of a diagnosis of PKS as well as for management and genetic counseling. Up to now, this is most extensive study on prenatal findings in PKS and authors should have a look at that and properly reformulate the sentence making a more detailed description of prenatal signs associated with PKS diagnosis

Introduction

Line 41 to 48: I would suggest to provide a more comprehensive and specific description of both PKS prenatal and postnatal typical features.

Discussion

Line 87-90: “The most frequent sonographic indicators are polyhydramnios (84%), congenital diaphragmatic hernia (16%) and micromelia mainly of rhizomelic type (10%). Other more sporadic anomalies are: fetal hydrops (6%), hygroma colli (3%), increased nuchal translucency (3%), macrosomia (3%), ventriculomegaly (3%), dilation of cavum pellucidum (3%), absence of stomach visualization (3%), and presence of sacral appendix (3%) (7)”.

Again, I would suggest to report updated data and percentage from most recently literature review (Am J Med Genet A. 2018 Dec;176(12):2575-2586. doi: 10.1002/ajmg.a.40499. Epub 2018 Oct 5).

Line 92-94: “There are, however, no pathognomonic echographic indicators. In our case, only polyhydramnios, ductus venous agenesis (DVA) and mild diastolic dysfunction were observed, while the fetal profile, biometry and the rest of the fetal parameters were normal…”

The presence of a fetal estimated weight above the mean (77 th percentile) in conjunction with polyhydramnios should alert for a possible overgrowth syndromic condition (especially if associated with other signs as DVA and/or in presence of mild cardiac abnormality) including also PKS. When PKS is prenatally suspected the laboratory should be advised in order to not use long-term cultures and/or in order to perform an a-CGH analysis. Also FISH analysis by using alpha-satellite or 12 fluorescent probes on interphase nuclei of amniotic cells usually increases the i(12p) detection rate. 

Authors might argue this point, comparing their findings with literature and reporting also, if available, all fetal growth parameters as PKS fetus usually present specific prenatal growth pattern(es. FL, BD). They can not say that DVA by itself is a marker for PKS, but they might suggest that its presence in conjunction with polyhdramnios and fetal weight above the mean or other minor signs should alert also for PKS.

Line 97-98: “Although prenatal diagnosis of PKS can be performed by chorionic villus sampling (CVS), amniocentesis or cordocentesis, amniocentesis seem to be the best method” Again, authors can find more detailed information about the prenatal diagnostic test detection rate in the previously suggested paper.

Author Response

REVIEWER #2: Major Comments: 

Comment 1.- Although the manuscript does not provide any specific new inside the prenatal cytogenetic diagnostic approach for PKS, it’s always useful reporting on a new sign associated with a such rare as well as likely underdiagnosed condition as PKS. I would suggest authors to provide a more specific background on prenatal PKS findings, carefully studying all the dedicated literature (there are some recent studies not even cited in the text and some data reported from literature are not properly updated). It would be interesting also to report a fetal post-mortem examination if an autopsy has been performed in order to get confirmation of prenatal findings detected by ultrasonography. Overall the manuscript is well-written, however it may benefit of some polishing by a native English collegue. 

 Answer to comment 1.- We thank the reviewer for his/her kind comments and recommendations. Unfortunately, termination of pregnancy was carried out in a different centre and no autopsy report was available. It has been reported in the new revised version of the manuscript. New references have been added as well as more details on prenatal and postnatal PKS findings. In addition, the language of the manuscript has been carefully revised by a native English Professor, following the recommendation of the reviewer. 

 Minor Comments: Comment 2.- Abstract: Line 19: “Pallister-Killian Syndrome (PKS) is a rare sporadic disorder with specific tissue mosaic distribution of an extra 12p isochromosome (i(12p)) that associates severe mental retardation as well as multiple and varied malformations”. I would say that the main clinical features of PKS are moderate to severe intellectual disability/neuromotor delay, skin pigmentation abnormalities, typical facial appearance variable associated with multiple congenital malformations and epilepsy 

 Answer to comment 2.- We fully agree with the reviewer's point. The paragraph has been corrected following his/her suggestion. “Its main clinical features are moderate to severe intellectual disability/neuromotor delay, skin pigmentation abnormalities, typical facial appearance, variable association with multiple congenital malformations and epilepsy”. 

 Comment 3.- Line 20: “Prenatal diagnosis is difficult as there are no associated identification signs ”Prenatal data from a cohort of 114 PKS probands have been recently published (Am J Med Genet A. 2018 Dec;176(12):2575-2586. doi: 10.1002/ajmg.a.40499. Epub 2018 Oct 5) providing guidelines for early recognition of the distinctive prenatal profile and consideration of a diagnosis of PKS as well as for management and genetic counseling. Up to now, this is most extensive study on prenatal findings in PKS and authors should have a look at that and properly reformulate the sentence making a more detailed description of prenatal signs associated with PKS diagnosis.

Answer to comment 3.- We thank the reviewer for his/her kind recommendation of this paper that we have read with interest. The text has been modified following according to the new insights reported: “Though prenatal findings (including congenital diaphragmatic hernia, ventriculomegaly, congenital heart disease, polyhydramnios, and rhizomelic shortening) have been described in literature, prenatal diagnosis is difficult as there are no distinctive or pathognomonic signs, and some of these malformations are hard to identify prenatally”. It has also been added to the bibliography list. 

Comment 4.- Introduction: Line 41 to 48: I would suggest to provide a more comprehensive and specific description of both PKS prenatal and postnatal typical features. 

Answer to comment 4.- Following the indication of the reviewer, this paragraph has been completed with more details about PKS prenatal and postnatal typical features. 

Comment 5.- Discussion: Line 87-90: “The most frequent sonographic indicators are polyhydramnios (84%), congenital diaphragmatic hernia (16%) and micromelia mainly of rhizomelic type (10%). Other more sporadic anomalies are: fetal hydrops (6%), hygroma colli (3%), increased nuchal translucency (3%), macrosomia (3%), ventriculomegaly (3%), dilation of cavum pellucidum (3%), absence of stomach visualization (3%), and presence of sacral appendix (3%) (7)”. Again, I would suggest to report updated data and percentage from most recently literature review (Am J Med Genet A. 2018 Dec;176(12):2575-2586. doi: 10.1002/ajmg.a.40499. Epub 2018 Oct 5).

Answer to comment 5.- Following the reviewer's recommendation, this paragraph has been revised and the percentages of anomalies associated with the syndrome have been updated according to the aforementioned study. 

Comment 6.- Line 92-94: “There are, however, no pathognomonic echographic indicators. In our case, only polyhydramnios, ductus venous agenesis (DVA) and mild diastolic dysfunction were observed, while the fetal profile, biometry and the rest of the fetal parameters were normal…” The presence of a fetal estimated weight above the mean (77 th percentile) in conjunction with polyhydramnios should alert for a possible overgrowth syndromic condition (especially if associated with other signs as DVA and/or in presence of mild cardiac abnormality) including also PKS. When PKS is prenatally suspected the laboratory should be advised in order to not use long-term cultures and/or in order to perform an a-CGH analysis. Also, FISH analysis by using alpha-satellite or 12 fluorescent probes on interphase nuclei of amniotic cells usually increases the i(12p) detection rate. Authors might argue this point, comparing their findings with literature and reporting also, if available, all fetal growth parameters as PKS fetus usually present specific prenatal growth pattern (es. FL, BD). They cannot say that DVA by itself is a marker for PKS, but they might suggest that its presence in conjunction with polyhdramnios and fetal weight above the mean or other minor signs should alert also for PKS. 

Answer to comment 6.- Following the editor's recommendation, this paragraph has been rewritten and updated according to the aforementioned study. Fetal biometry report has been added in the figure 1 and a paragraph was also included in the discussion section of the revised version of the manuscript on fetal biometry parameters. 

Comment 7.- Line 97-98: “Although prenatal diagnosis of PKS can be performed by chorionic villus sampling (CVS), amniocentesis or cordocentesis, amniocentesis seem to be the best method” Again, authors can find more detailed information about the prenatal diagnostic test detection rate in the previously suggested paper. 

Answer to comment 7.- Following the reviewer's comment, this paragraph has been corrected as follows: Although prenatal diagnosis of PKS can be performed by chorionic villus sampling (CVS), amniocentesis or cordocentesis, amniocentesis seem to be the best method with detection rates reported ranging from 40 to 75% for CVS and 78-95% for amniocentesis. Due to the high risks associated to cordocentesis, it is not a first option diagnostic choice.
